# Analysis of Beamforming Antenna for Practical Indoor Location-Tracking Application

**DOI:** 10.3390/s19143040

**Published:** 2019-07-10

**Authors:** Sun-Woong Kim, Dong-You Choi

**Affiliations:** 1IT Research Institute, Chosun University, Gwangju 61452, Korea; 2Department of Information and Communication Engineering, Chosun University, Gwangju 61452, Korea

**Keywords:** beamforming antenna, Butler matrix, tapered slot antenna, indoor location tracking, IR-UWB radar

## Abstract

The single antenna used in conventional ultra-wideband radar has difficulty tracking targets over a wide range because of a relatively narrow beamwidth. Herein, we propose a beamforming antenna that can track targets over a wide range by electronically controlling the main beam of the antenna. The proposed beamforming antenna was fabricated by connecting a 1 × 4 linear array antenna and a 4 × 4 Butler matrix. The Butler matrix was fabricated in a laminated substrate using two TRF-45 substrates. Furthermore, the input Ports 1–4 generate a phase difference at regular intervals in each output port, and the output phase is fed to the array antenna. The proposed tapered-slot antenna was fabricated on a Taconic TLY substrate, and the impedance bandwidth of the antenna was achieved within a wide bandwidth of 4.32 GHz by satisfying a VSWR (Voltage Standing Wave Ratio) ≤2 within the 1.45–5.78 GHz band. Furthermore, the fabricated antenna has directional radiation patterns, which was found to be a suitable characsteristic for location tracking in a certain direction. Finally, the beamforming antenna has four beamforming angles, and to verify the performance for an indoor location tracking application of impulse-radio ultra-wideband radar, it was connected to an NVA-R661 module.

## 1. Introduction

The field of indoor location tracking has recently been applied to various fields, such as military, medical, and civilian areas. Among them, an IR-UWB (Impulse Radio Ultra-Wideband) radar system is a typical example of an indoor location tracking system and makes precise location tracking possible [1]. 

The antenna used in an IR-UWB radar system must have a wide bandwidth to transmit and receive the impulse signals of nano-second units in a specific direction, as well as high gain and directional radiation patterns to track the location of an object. Conventional IR-UWB antennas have been studied for various uses, such as in Vivaldi antennas [2,3], patch antennas [4], Yagi-type antennas [5,6], and tapered-slot antennas (TSA) [7,8]. These antennas have difficulty tracking objects that are located over a wide area owing to the relatively narrow beamwidth. Furthermore, the individual studies on antennas and their applications in conventional indoor location tracking have not considered the needs of other antenna types. 

In this paper, an indoor location-tracking technique based a beamforming antenna developed through a linkage study is proposed.

The proposed beamforming antenna was fabricated by combining beamforming networks that generate a constant phase with a tapered-slot antenna array. The structure of the beamforming antenna requires a beamforming network (N × N Butler matrix) capable of providing different constant phases, and an N-array antenna to radiate toward the desired direction [9,10]. The proposed technology was implemented using a Butler matrix, which is easy to manufacture at low cost. The Butler matrix consists of 3-dB/90° slot-directional hybrid couplers and 45° phase shifters. When the input and output ports of the Butler matrix increase, the controlled phase beam increases; however, the circuit size also increases, and the design becomes complicated. Therefore, we propose a 4 × 4 Butler matrix with four input and four output ports, where the output phase is fed into the input ports of the linear array antenna. Finally, the proposed beamforming antenna verifies the applicability of indoor location tracking by connecting an NVA-R661 module for a conventional IR-UWB radar.

The remainder of this paper is organized as follows. In Section 2, the fabrication of the proposed beamforming antenna for indoor location tracking is described and analyzed. In Section 3, to verify the performance of indoor location tracking, the fabricated beamforming antenna is connected to the NVA-R661 module for a IR-UWB radar. In Section 4, some concluding remarks are provided.

## 2. Antenna Design and Analysis

The proposed beamforming antenna for an indoor wireless positioning system is shown in Figure 1. The proposed beamforming antenna was designed by connecting a 1 × 4 array antenna and a 4 × 4 Butler matrix. When configuring the arrays, the distance between each antenna is an extremely important parameter that determines the direction and gain of the main beam. Grating lobes are generated when the distance between each antenna is larger than the wavelength. In contrast, if the distance between each antenna is small, the side lobe level increases owing to the interference between antennas. Therefore, the distance d of the array antenna must satisfy the condition *λ*/2 < *d* < *λ*, and a distance d of 30 mm was chosen.

### 2.1. Design and Measurement of Tapered Slot Antenna (TSA)

The configuration of the proposed tapered slot antenna is shown in Figure 2. As Figure 2 indicates, the radiation element is placed on the top layer of the proposed TSA, whereas the transition feed is placed on the bottom layer. The directivity and beamwidth of the proposed TSA can be expressed as (1) and (2).
$$\mathrm{Directivity}\;\left[\mathrm{dB}\right]=10\log\left(10\frac{L_{T}}{\lambda_{0}}\right),$$
(1)$$\mathrm{Beamwidth}\;\left[{}^{\circ}\right]=\;\frac{55}{\sqrt{L_{T}/\lambda_{0},}},$$
$$\mathrm{Directivity}\;\left[\mathrm{dB}\right]=10\log\left(4\frac{L_{T}}{\lambda_{0}}\right),$$
(2)$$\mathrm{Beamwidth}\;\left[{}^{\circ}\right]=\;\frac{77}{\sqrt{L_{T}/\lambda_{0},}},$$
where *λ*_0_ is the wavelength for the center frequency of the proposed TSA, and *L_T_* is the aperture length [11]. The proposed tapered slot antenna was designed using HFSS ver. 12 (Ansys Co., Canonsburg, PA, USA), and the design parameters were determined through a repeated analysis. Furthermore, the antenna was fabricated through an etching process using a Taconic TLY substrate, which has a relative permittivity of 2.2, loss tangent of 0.0009, and thickness of 1.52 mm.

The design parameters of the proposed tapered slot antenna were as follows: *L_S_* = 140 mm, *W_S_* = 90 mm, *L_T_* = 96.8 mm, *L_T_*__*1*_ = 23 mm, *W_T_* = 80 mm, *L_m_* = 45 mm, *L_m_*__*1*_ = 15 mm, and *W_m_* = 5 mm.

The impedance bandwidth and gain results of the fabricated tapered slot antenna are shown in Figure 3.

The simulation results in Figure 3 show that the impedance bandwidth of the fabricated tapered slot antenna achieved a wide bandwidth of 4.29 GHz, which was satisfied by VSWR ≤ 2 within the 1.45–5.74 GHz band. The simulation gain results are 6.7, 8.36, and 8.97 dBi at 3–5 GHz, respectively. The measurement results show that impedance bandwidth of the fabricated TSA achieves a wide bandwidth of 4.32 GHz by satisfying VSWR ≤ 2 within the 1.46–5.78 GHz band. The measurement gain results are 6.17, 8.19, and 9.18 dBi at 3–5 GHz, respectively.

The radiation pattern results of the fabricated tapered slot antenna were analyzed in the E-plane (y-z) and H-plane (x-z), as shown in Figure 4.

The results in Figure 4 indicate that the E-plane and H-plane radiation patterns of the proposed TSA exhibit the directivity of an end-fire, which increases the sensitivity for a certain direction. The 3 dB beamwidth simulation results of the E-plane and H-plane were 70.45° and 117.00° at 3 GHz, 58.03° and 96.38° at 4 GHz, and 36.48° and 72.25° at 5 GHz. Furthermore, the measured results show 72° and 109° at 3 GHz, 40° and 84° at 4 GHz, and 29° and 60° at 5 GHz, respectively.

Comparisons of the simulation and measurement results of the fabricated TSA are shown in Table 1.

As shown in Table 1, the simulation and measurement results of the fabricated TSA show a proper agreement, although small errors were observed in certain areas. The reason for this is presumably due to errors during the fabrication process and a loss between the antenna and a connector.

### 2.2. Design and Measurement of Butler Matrix

The proposed 4 × 4 Butler matrix consists of 3-dB/90° slot-directional hybrid couplers and 45° phase shifters, and has four input/output ports. The proposed 4 × 4 Butler matrix is shown in Figure 5. The 3-dB/90° slot-directional hybrid coupler and 45° phase shifter are operated through the coupling slot between the coupling patches located at the top and bottom layers. The proposed 3-dB/90° slot-directional hybrid couplers and 45° phase shifter can be expressed as (3).
(3)$$L_{p}\left(L_{g}\right)=\;\frac{\lambda_{c}}{4}\;{\left[1-{\left(\frac{\pi \left(W_{p}+W_{g}\right)}{4\lambda_{c}}\right)}^{2}\right]}^{-1},t$$
where the length *L_p_* (*L_g_*) of the coupling patch and coupling slot can be calculated from the effective wavelength λ_c_ at the center frequency of the coupler, and from the shifter and width *W_p_*, *W_G_* of the coupler patch and coupler slot [9]. The proposed 3-dB/90° slot-directional hybrid coupler and 45° phase shifter were designed using HFSS ver. 12, and the design parameters were determined through a repeated analysis.

As shown in Figure 5b, the proposed 3-dB/90° slot-directional hybrid couplers were designed as a laminated substrate using two TRF-45 substrates, which have a relative permittivity of 4.5, a loss tangent of 0.0035, and a thickness of 0.61 mm. The proposed 3-dB/90° slot-directional hybrid coupler consists of four ports, namely an input port, a direct port, a coupled port, and an isolated port, and is placed on the two signal lines on the top and bottom layers. The ground plane of the intermediate layer is inserted into a rectangular slot, and its structure is mutually coupled. The design parameters of the 3-dB/90° slot-directional hybrid coupler are as follows: *L_S_* = 30 mm, *W_S_* = 12.8 mm, *L_p_* = 10 mm, *W_p_* = 5 mm, *W_p1_* = 3 mm, *L_g_* = 10 mm, *Wg* = 6.4, and *W_m_* = 1.2 mm.

For the proposed 45° phase shifter shown in Figure 5c, the signal lines were placed on the top and bottom layers. The ground plane of the intermediate layer was inserted into a rectangular slot, and its structure was mutually coupled. The design parameters of the 45° phase shifter are as follows: *L_S_* = 43.6 mm, *W_S_* = 25 mm, *L_p_* = 9 mm, *W_p_* = 5 mm, *W_p1_* = 3.2 mm, *L_g_* = 9 mm, *W_g_* = 6.6 mm, *L_m_* = 13.1 mm, *L_m1_* = 10.4, and *W_m_* = 1.2 mm.

The S-parameters of the proposed 3-dB/90° slot-directional hybrid coupler were analyzed in terms of the return loss S11, isolation S41, and insertion losses S21, S31, which are shown in Figure 6.

The simulation results in Figure 6 show that the return loss S_11_ and isolation loss S_41_ achieved good results of lower than −20 dB within the 3–5 GHz band. In addition, the insertion losses S_21_ and S_31_ were 3 dB within the 3–5 GHz band; overall, the error observed was ±0.4 dB.

The results of the phase and phase difference simulation analyses of the proposed 3-dB/90° slot-directional hybrid coupler are shown in Figure 7.

The simulation results in Figure 7 show that the S_31_ and S_21_ phases of the proposed 3-dB/90° slot-directional hybrid coupler are 227° and 138° at 3 GHz, 161° and 72° at 4 GHz, and 95° and 5° at 5 GHz. The simulation analysis of the phase difference shows an error of ±2° at 90°.

The S-parameter simulation results of the proposed 45° phase shifter were analyzed in terms of the return loss S_11_ and insertion loss S_21_, which are indicated in Figure 8.

The simulation results in Figure 8 show that the return loss S_11_ achieved good results of less than −20 dB within the 3–5 GHz band. The insertion loss S_21_ was between −0.4 and −0.7 dB within the 3–5 GHz band.

The results of the phase and phase difference simulation analyses of the proposed 45° phase shifter are shown in Figure 9.

The simulation results in Figure 9 indicate that the S_21_ and S_31_ phases of the proposed 45° phase shifter are 171° and 125° at 3 GHz, 91° and 46° at 4 GHz, and 12° and −32° at 5 GHz. The phase difference simulation analysis showed an error of ±2° at 45°.

Therefore, the return loss results of the proposed coupler and shifter are extremely low, with good insertion loss and isolation characteristics. Furthermore, the phase difference of the coupler and shifter was approximately 90° and 45° within the proposed bandwidth, respectively. The proposed Butler matrix is realized by combining the designed four couplers and two shifters, and the design parameters of the Butler matrix are the same as those of the couplers and shifters.

The S-parameter simulation results of the Butler matrix for input Ports 1 and 2 are shown in Figure 10 and Figure 11, respectively.

The simulation results in Figure 10 show that the return loss S_11_ and isolation losses S_21_, S_31_, and S_41_ are less than −20 dB within the proposed bandwidth. The insertion losses S_51_, S_61_, S_71_, and S_81_ are low at approximately 6–6.7 dB within the proposed bandwidth.

The simulation results in Figure 11 show that the return loss S_22_ and isolation losses S_12_, S_32_, and S_42_ are less than −20 dB within the proposed bandwidth. The insertion losses S_52_, S_62_, S_72_, and S_82_ are low at approximately 6–7.4 dB within the proposed bandwidth.

The results of the phase and phase difference simulation analyses for Ports 1 and 2 of the proposed Butler matrix are shown in Figure 12, Figure 13, Figure 14 and Figure 15, and are listed in Table 2.

The simulation results for Port 1 in Table 2 show that, within the 3 GHz band, the phase is 17° and 59° for P_51_ and P_61_, 59° and 103° for S_61_ and S_71_, and 103° and 148° for S_71_ and S_81_, respectively. The phase for Port 2 is 103° and −29° for P_52_ and P_62_, −29° and −167° for S_62_ and S_72_, and −167° and 58° for S_72_ and S_82_, respectively. Therefore, the simulation results of the phase difference for Ports 1 and 2 are −45° ± 3° and 135 ± 2° within the 3 GHz band. Similar results are shown at the other frequency bands (4 and 5 GHz).

The S-parameter measurement results of the Butler matrix for input Ports 1 and 2 are shown in Figure 16 and Figure 17.

The measurement results in Figure 16, show that the return loss S_11_ and isolation losses S_21_, S_31_, and S_41_ were less than −15 dB within the 3–5 GHz band. The insertion losses S_51_, S_61_, S_71_, and S_81_ showed a low insertion loss of approximately 6.2–6.8 dB within the proposed bandwidth.

The measurement results in Figure 17 show that the return loss S_22_ and isolation losses S_12_, S_32_, and S_42_ are less than −10 dB within the 3–5 GHz band. In addition, the insertion losses S_52_, S_62_, S_72_, and S_82_ show a low insertion loss of approximately 6.1–6.7 dB within the proposed bandwidth.

The results of the phase and phase difference measurement analysis for Ports 1 and 2 of the proposed Butler matrix are shown in Figure 18, Figure 19, Figure 20 and Figure 21, and are listed in Table 3.

The measurement results for Port 1 in Table 3 show that within the 3 GHz band, the phase is −68° and −25° for P_51_ and P_61_, −25° and 20° for S_61_ and S_71_, and 20° and 62° for S_71_ and S_81_. The phase of Port 2 is 19° and −116° for P_52_ and P_62_, −116° and 109° for S_62_ and S_72_, and 109° and −30° for S_72_ and S_82_. Therefore, the measurement results of the phase difference for Ports 1 and 2 are −45° ± 3° and 135 ± 4° at 3 GHz. Similar results are shown at the other frequency bands (4 and 5 GHz). The simulation and measurement analysis results of the fabricated Butler matrix in Table 2 and Table 3 show similar results at 3, 4, and 5 GHz. In addition, Ports 3 and 4 are symmetrical with respect to Ports 1 and 2, and output opposite polarities of −135° and 45°, respectively.

In order to observe the characteristic for the time delay of the proposed 4 × 4 Butler matrix in a wide bandwidth, the group delay is analyzed and is shown in Figure 22. 

In the group delay analysis in Figure 22, the results of output ports for input Ports 1 and 2 has a constant time delay characteristic at about 60 nano second.

Therefore, the output phase from the Butler matrix is fed into the input port of the 1 × 4 array antenna. A photograph of the fabricated Butler matrix is shown in Figure 23.

The simulation results of the radiation pattern for the proposed beamforming antenna are shown in Figure 24. 

The results in Figure 24 indicate that the simulated beamforming angles at 3 GHz are −22°, +60°, −62°, and +22°. At 4 GHz, the simulation results are −16°, +45°, −50°, and +16°. At 5 GHz, the simulation results are −14°, +38°, −38°, and +14°. The simulation gain results at each beamforming angle are 7.26, 11.06, 11.04, and 6.89 dBi within the 3 GHz band; 10.88, 13.39, 13.27, and 10.73 dBi within the 4 GHz band; and 12.08, 13.63, 13.77, and 12.71 dBi within the 5 GHz band, respectively.

The radiation patterns of the fabricated beamforming antenna were measured using a far-field analysis system in an anechoic chamber room (EMTI Co., Seoul, Korea), as shown in Figure 25.

When the feed wave was fed to each input port, the terminating resistance was connected to the remaining ports. The results of the radiation pattern analysis of the fabricated beamforming antenna are shown in Figure 26.

The results in Figure 26 show that the measured beamforming angles are −21°, +59°, −56°, and +23° within the 3 GHz band; −14°, +46°, −44°, and +16° within the 4 GHz band; and −13°, +44°, −36°, and +17° within the 5 GHz band, respectively. The measurement gain results at each beamforming angle are 7.98, 10.40, 10.04, and 8.01 dBi at 3 GHz; 9.25, 11.41, 10.98, and 9.41 dBi at 4 GHz, and 9.17, 10.05, 9.10, and 10.07 dBi at 5 GHz, respectively.

The proposed beamforming antenna shows similar results within the proposed frequency band. Therefore, the maximum beamforming range of the fabricated antenna is 115° (+59° to −56°) within the 3 GHz band, 90° (+46° to −44°) within the 4 GHz band, and 80° (+44 to −36°) within the 5 GHz band.

## 3. Target Tracking Test of Fabricated Beamforming Antenna 

This chapter describes a simple experiment conducted to verify the practicability of the fabricated antenna. The configuration of the experiment is shown in Figure 27.

As shown in Figure 27, the proposed radar system is connected to the fabricated beamforming antenna and NVA-R661 UWB radar module (Xethru Co., Oslo, Norway) [12]. The proposed radar system has four measurement regions according to the four input ports, and the angle *θ* between each point is 15°.

### Test of Indoor Location Tracking for Beamforming Antenna

The radar and signal processing procedure is shown in Figure 28.

As shown in Figure 28, the signal processing procedure consists of clutter reduction, detection, and localization and tracking steps. The raw data detected at each port are signal processed using the NVA-R661 module. The raw data reflected from a single target are given by:(4)$$R_{i}=R_{t,i}+R_{c,i}+\;R_{n,i},$$
where *R_i_* is the *i*-th raw data reflected from a single target and consists of the target signal *R_t,i_*, clutter signal *R_c,i_*, and noise *R_n,i_*. In the clutter reduction step, the main purpose is to remove the clutter signal included in the raw data. To remove the clutter signal, the singular value decomposition algorithm is used. The *n* signal reflected comprises matrix *R* with *R* = [*R*_1_, *R*_2_, …, *R*_n_]*^t^*. Matrix *R* is given by:(5)$$R=R_{T}+R_{C}+\;N$$

Matrix *R* consists of the target signal *R_T_*, clutter signal *R_C_*, and noise *N*. To remove the clutter signal in *R*, it is decomposed using *R* = *USV^T^*, where *R* is an m × n matrix, and *U* and *V* are the orthogonal matrices of *m* × *m* and *n* × *n*. Therefore, *R* is given by:(6)$$R=\;\overset{j}{\underset{i=1}{\sum }}\sigma_{i}u_{i}v_{i}{}^{T}$$
where *u_i_* is the *i*-th singular value of *R*, and *σ_i_* and *vi* are the *i*-th eigenvectors to the left and right of *R*. The target *R_T_* is obtained using *R* and *R_C_*.
(7)$$R_{T}+N=R-R_{C}$$

During the detection step, the location of the target is determined [13,14,15,16]. Finally, during the localization and tracking step, the distance of a target is detected using the time of arrival [17,18]. 

A single target analysis using the proposed beamforming antenna is shown in Figure 29.

The results in Figure 29 show that at Port 3, a target is detected at 2 m within the regions of P2, P3, and P4. At Port 1, a target is detected at 2 m within the regions of P4, P5, and P6. At Port 4, a target is detected at 2 m within the regions of P6, P7, and P8. At Port 2, a target is detected at 2 m within the regions of P8, P9, and P10. Therefore, the results of the target tracking show that the target is detected at 2 m within the region of P2–P10.

The proposed 4 × 4 beamforming antenna for the indoor location tracking system was compared with antennas applied in similar systems, as listed in Table 4.

As shown in Table 4. the beamforming antenna in [9] led to a great deal of inspiration for this work, but it somewhat lacks the measurement results. In the case of [18] and [19], it provides an indoor location tracking system in 1 dimension and 2 dimensions coordinate system. In Ref. [19], the indoor location tracking measurement of 2 dimension coordinate require two radars to operate. The proposed 4 × 4 beamforming antenna has the advantage of measuring the targets located at various angles. Therefore, the fabricated beamforming antenna provides a solution to enable the tracking of targets over a wide range compared with a conventional IR-UWB antenna.

## 4. Conclusions

In this paper, an overview of a beamforming antenna that can track the location of a target over a wide range was described. Conventional IR-UWB antennas have difficultly tracking a target over a wide range because of the relatively narrow beamwidth. To solve this problem, a beamforming antenna that tracks a target over a wide range by electronically controlling the beam of the antenna was proposed.

The beamforming antenna was fabricated by connecting a 4 × 4 Butler matrix and a 1 × 4 array antenna. The feed wave generated in the 4 × 4 Butler matrix is fed into the 1 × 4 array antenna, which controls the four beams.

The fabricated beamforming antenna has beamforming angles of −21°, +59°, −56°, and +23° at 3 GHz; −14°, +46°, -44°, and +16° at 4 GHz; and −13°, +44°, −36°, and +17° at 5 GHz. Therefore, the beamforming angles of the fabricated beamforming antenna have maximum values of 115°, 90°, and 80° within the 3, 4, and 5 GHz bands, respectively. The results of location tracking show that a target can be detected at 2 m within the regions of P2–P10.

Previously published studies in this regard have not been able to meet the needs of individual antennas and applications. An IR-UWB antenna cannot track a broad range of targets owing to the narrow beamwidth of the directional pattern. In this study, we propose an antenna based on beamforming and verify the possibility of indoor location tracking technology within a wide range by applying a commercialized module. In addition, because the main beam of the antenna is electronically steered, it is possible to detect and track multiple targets at the same time such that they can be utilized in various application technologies. The development of a new indoor location tracking algorithm based on a beam steering system is, therefore, expected.

## Figures and Tables

**Figure 1 sensors-19-03040-f001:**
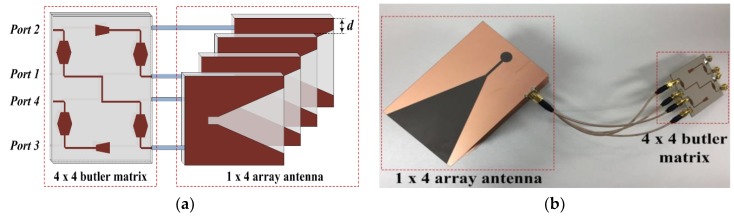
Configuration of the proposed beamforming antenna for indoor wireless positioning: (**a**) geometry and (**b**) photograph of antenna.

**Figure 2 sensors-19-03040-f002:**
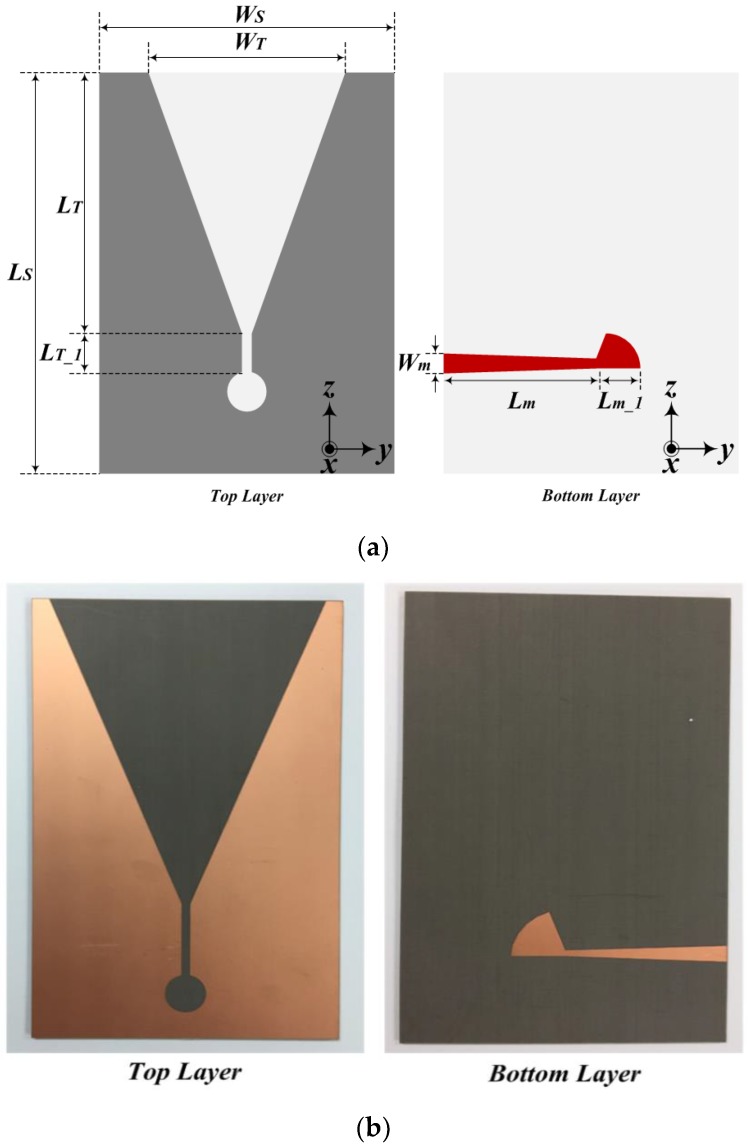
Configuration of the proposed tapered slot antenna: (**a**) geometry and (**b**) photograph of antenna.

**Figure 3 sensors-19-03040-f003:**
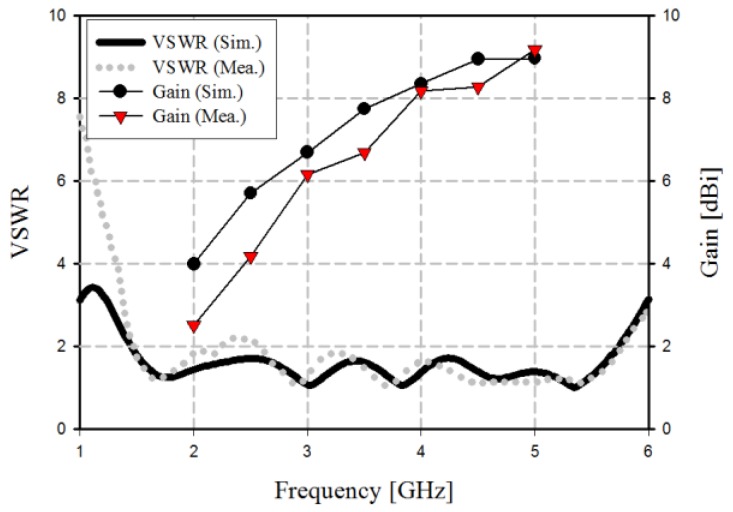
Results of the impedance bandwidth.

**Figure 4 sensors-19-03040-f004:**
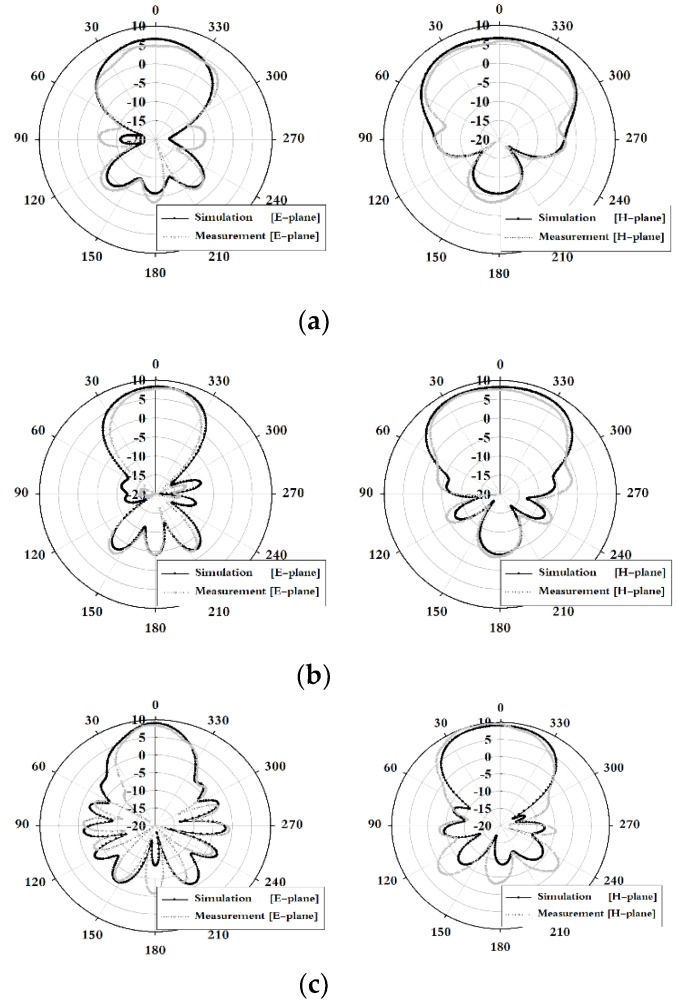
Simulation and measurement results of the radiation pattern: (**a**) 3, (**b**) 4, and (**c**) 5 GHz.

**Figure 5 sensors-19-03040-f005:**
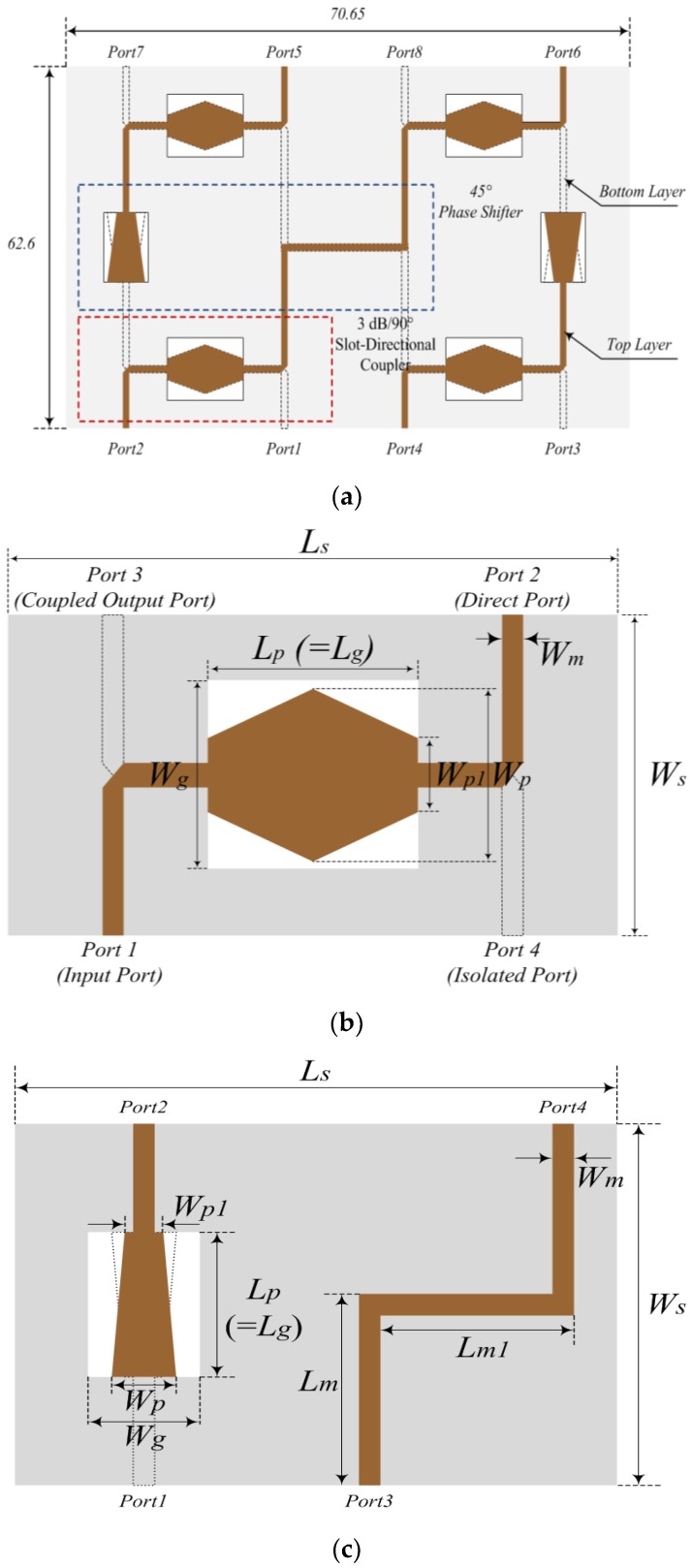
Configuration of the Butler matrix: (**a**) structure of the Butler matrix, (**b**) structure of the 3-dB/90° slot-directional hybrid coupler, and (**c**) structure of 45° phase shifter.

**Figure 6 sensors-19-03040-f006:**
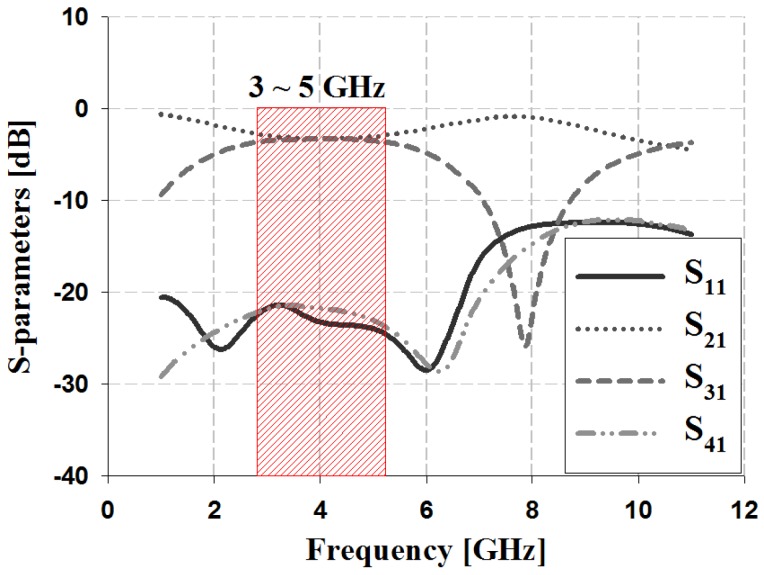
S-parameter simulation analysis of the proposed coupler.

**Figure 7 sensors-19-03040-f007:**
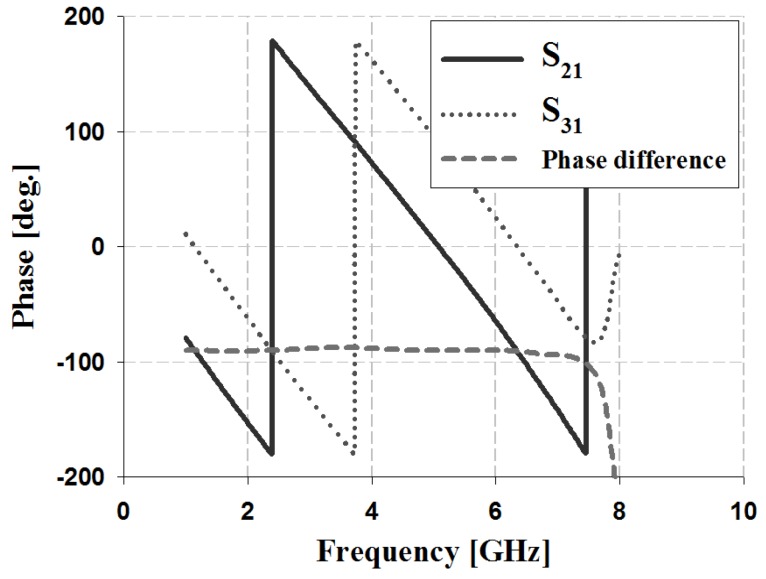
Phase and phase difference simulation analysis of the proposed coupler.

**Figure 8 sensors-19-03040-f008:**
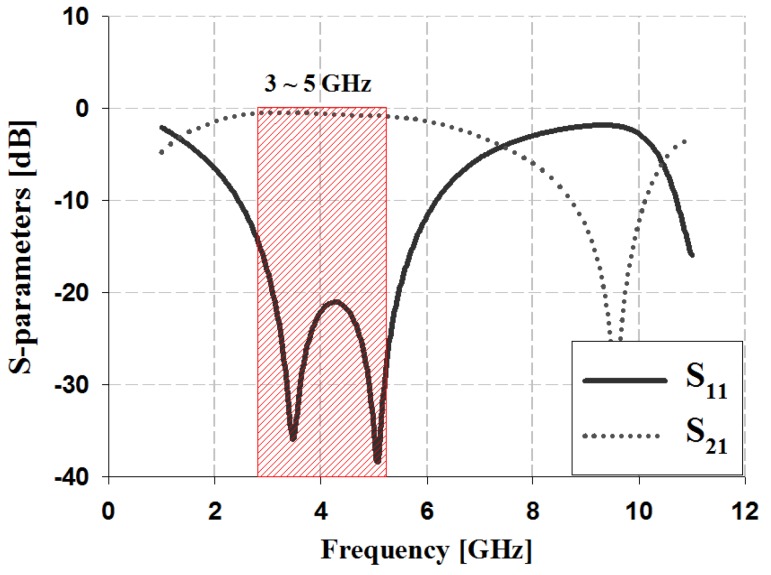
S-parameter simulation analysis of the proposed shifter.

**Figure 9 sensors-19-03040-f009:**
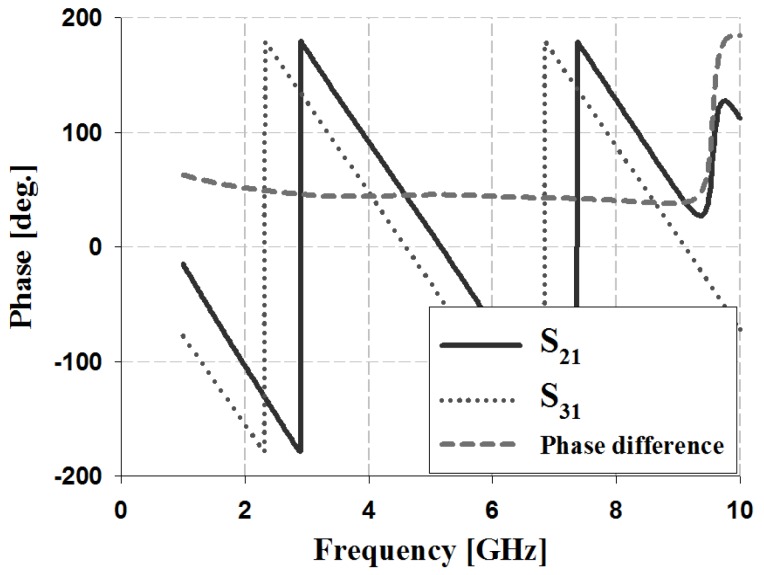
Phase and phase difference simulation analysis of the proposed shifter.

**Figure 10 sensors-19-03040-f010:**
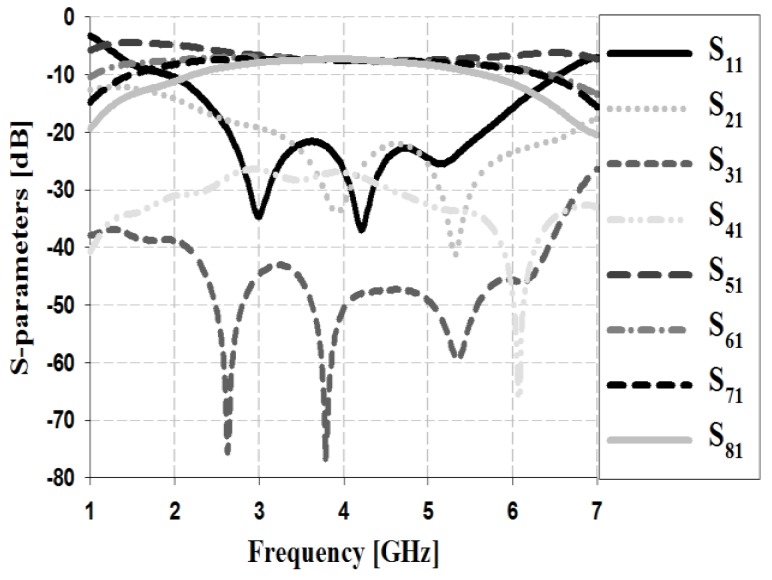
Simulation analysis of S-parameters for input Port 1.

**Figure 11 sensors-19-03040-f011:**
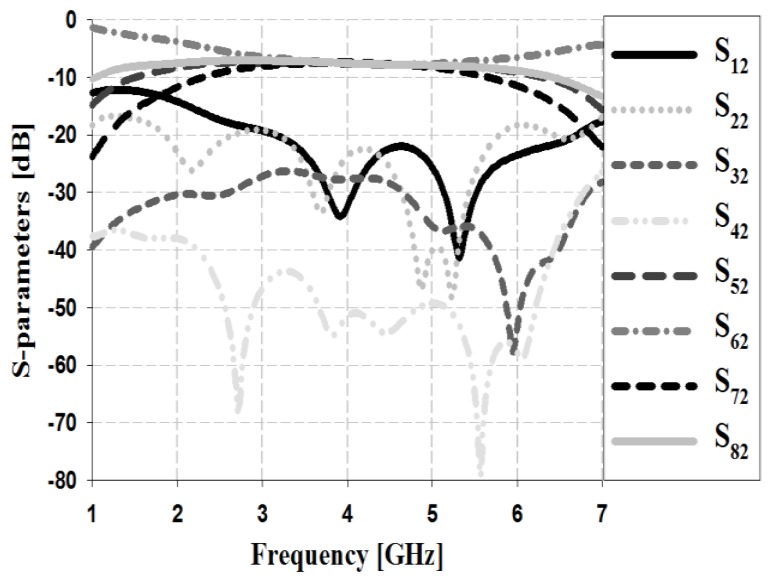
Simulation analysis of S-parameters for input Port 2.

**Figure 12 sensors-19-03040-f012:**
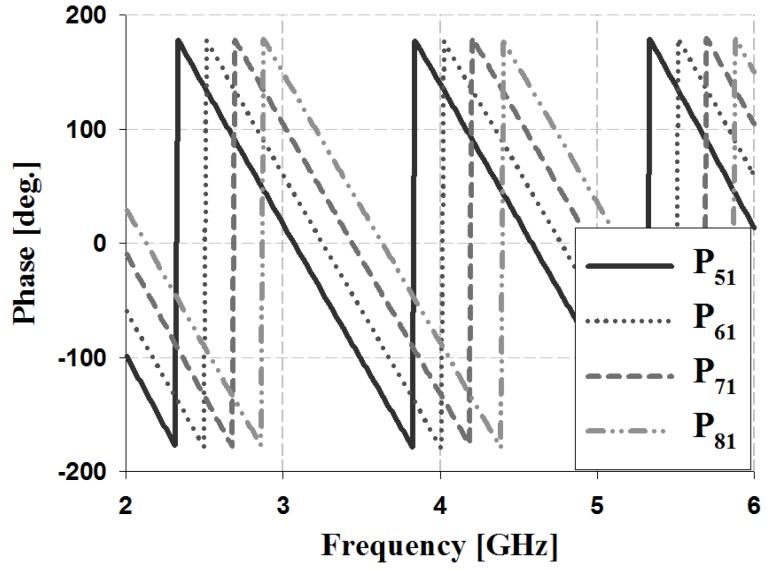
Simulation analysis of phase for input Port 1.

**Figure 13 sensors-19-03040-f013:**
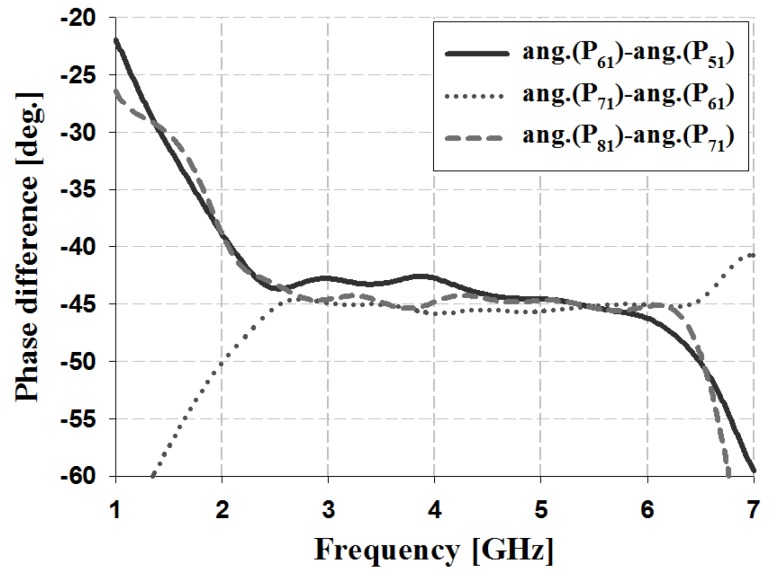
Simulation analysis of phase difference for input Port 1.

**Figure 14 sensors-19-03040-f014:**
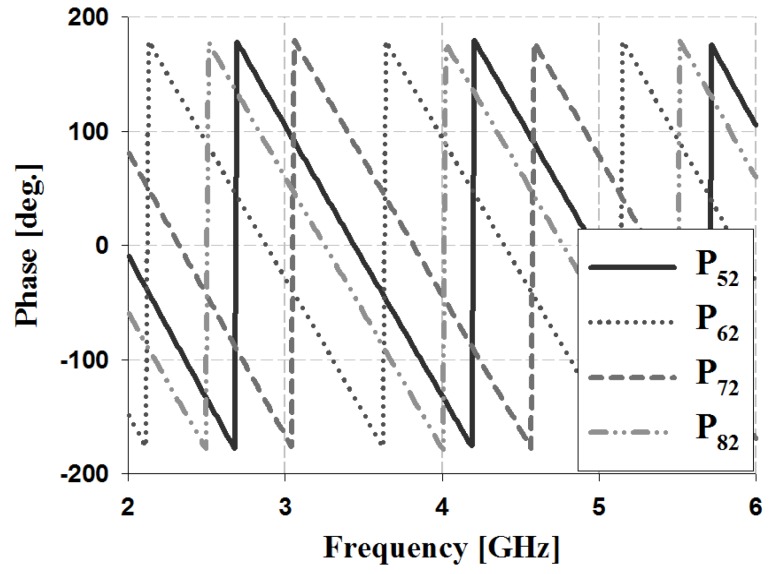
Simulation analysis of phase for input Port 2.

**Figure 15 sensors-19-03040-f015:**
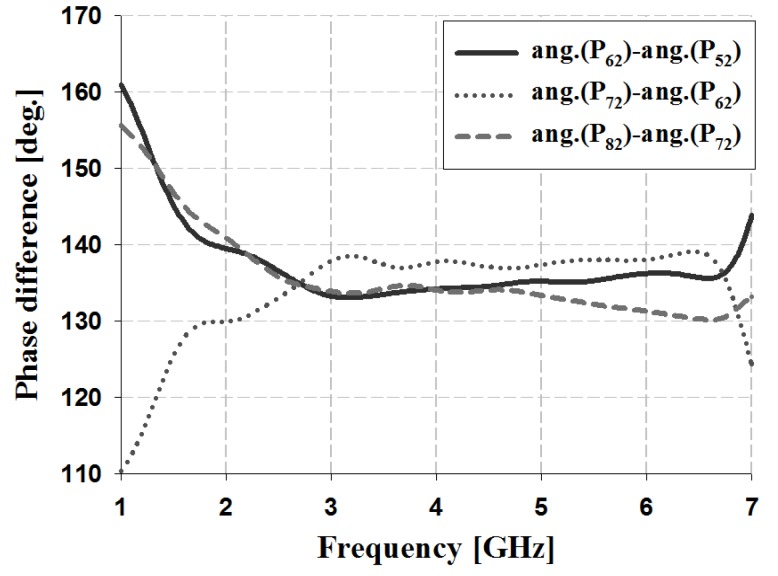
Simulation analysis of phase difference for input Port 2.

**Figure 16 sensors-19-03040-f016:**
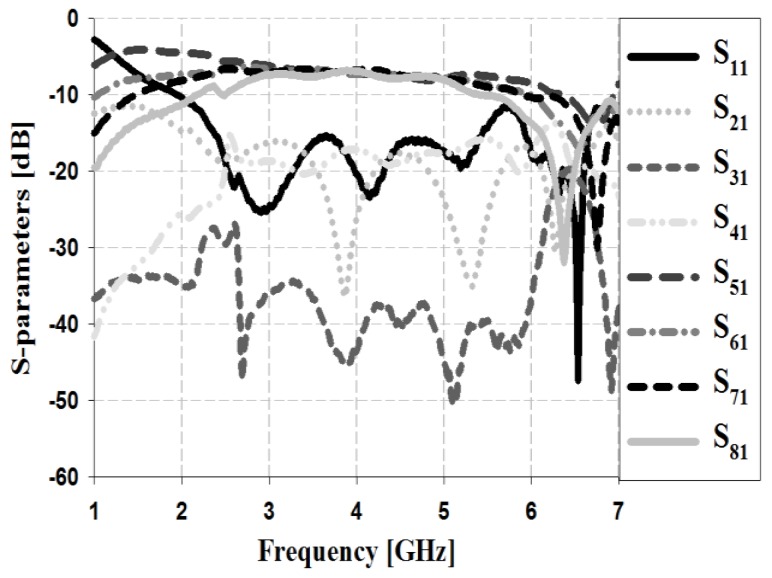
Measurement analysis of S-parameters for input Port 1.

**Figure 17 sensors-19-03040-f017:**
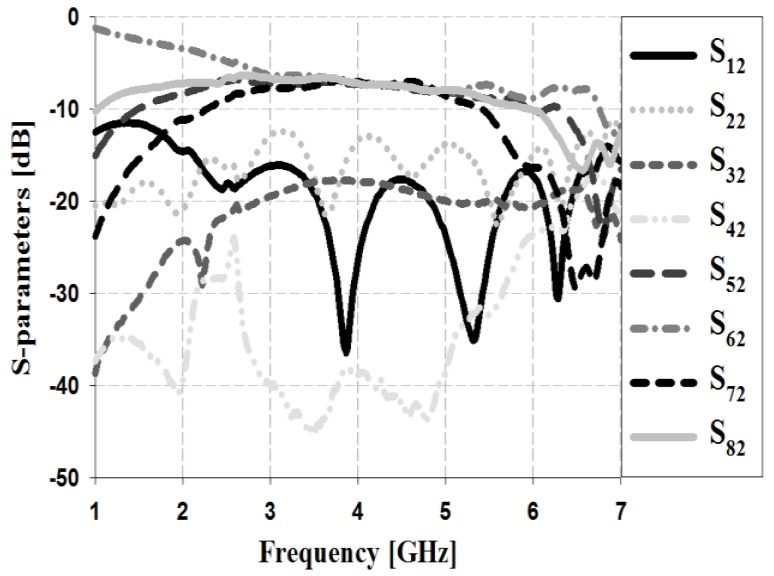
Measurement analysis of S-parameters for input Port 2.

**Figure 18 sensors-19-03040-f018:**
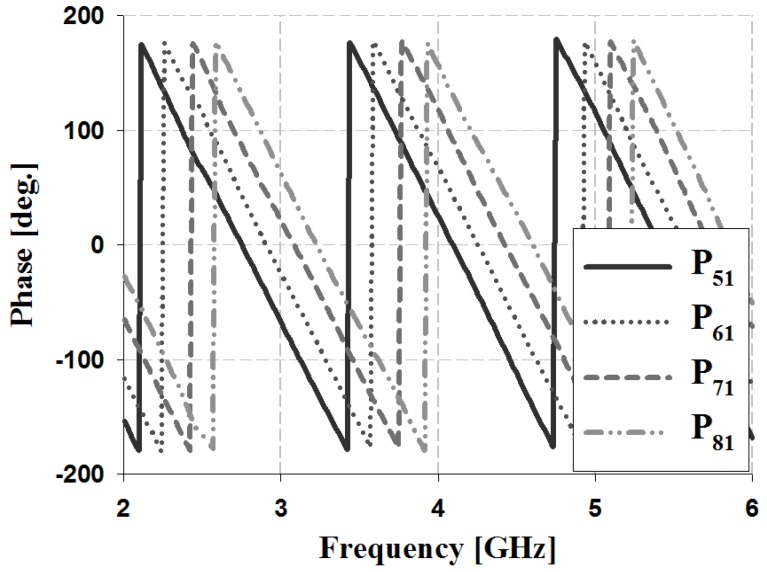
Measurement analysis of phase for input Port 1.

**Figure 19 sensors-19-03040-f019:**
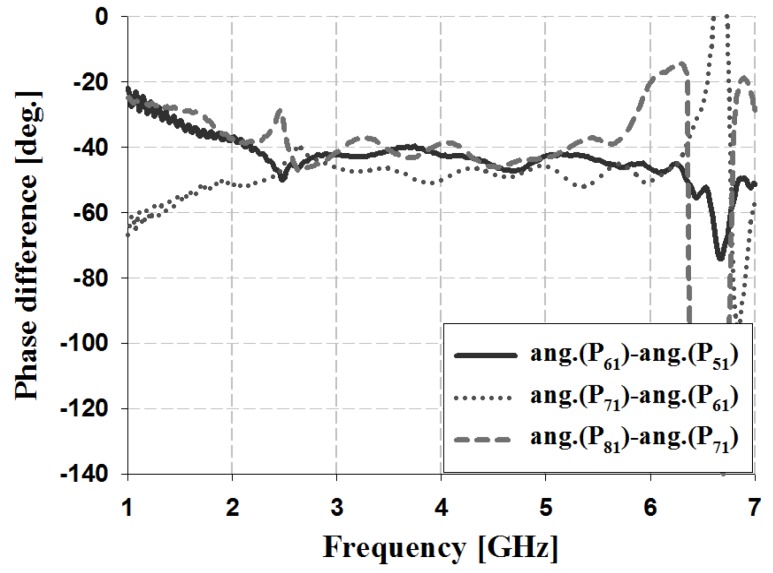
Measurement analysis of phase difference for input Port 1.

**Figure 20 sensors-19-03040-f020:**
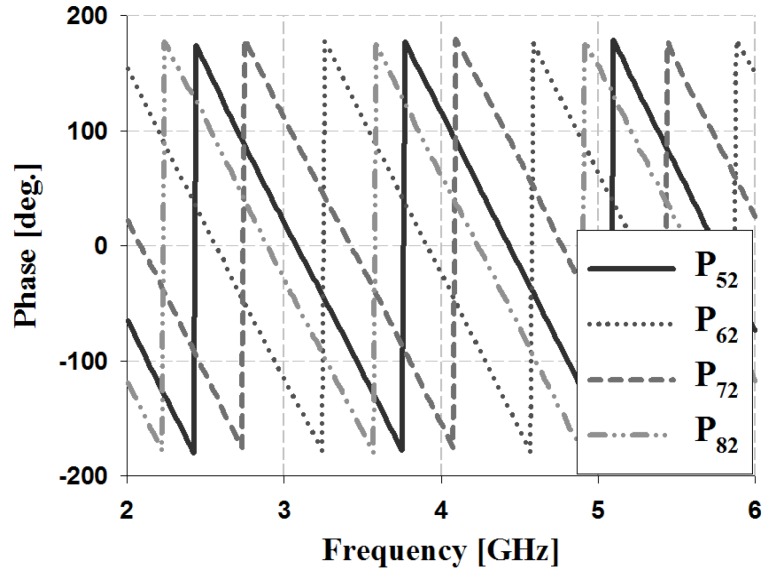
Measurement analysis of phase for input Port 2.

**Figure 21 sensors-19-03040-f021:**
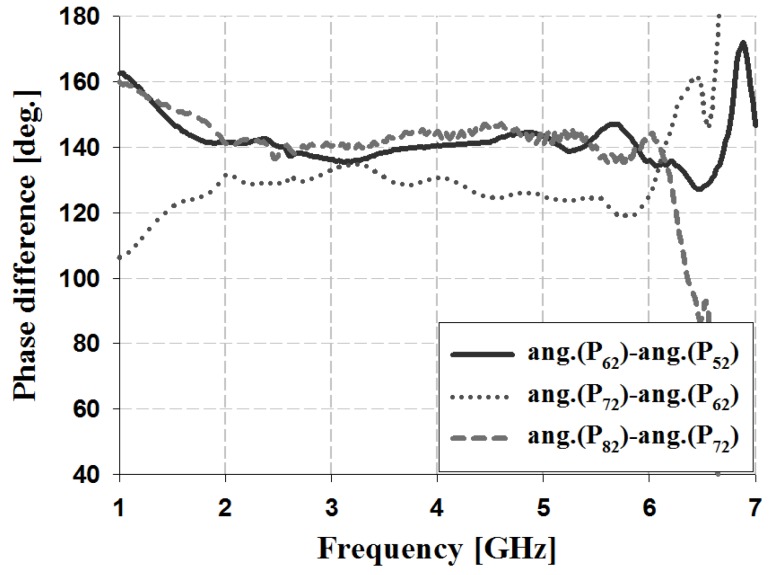
Measurement analysis of phase difference for input Port 2.

**Figure 22 sensors-19-03040-f022:**
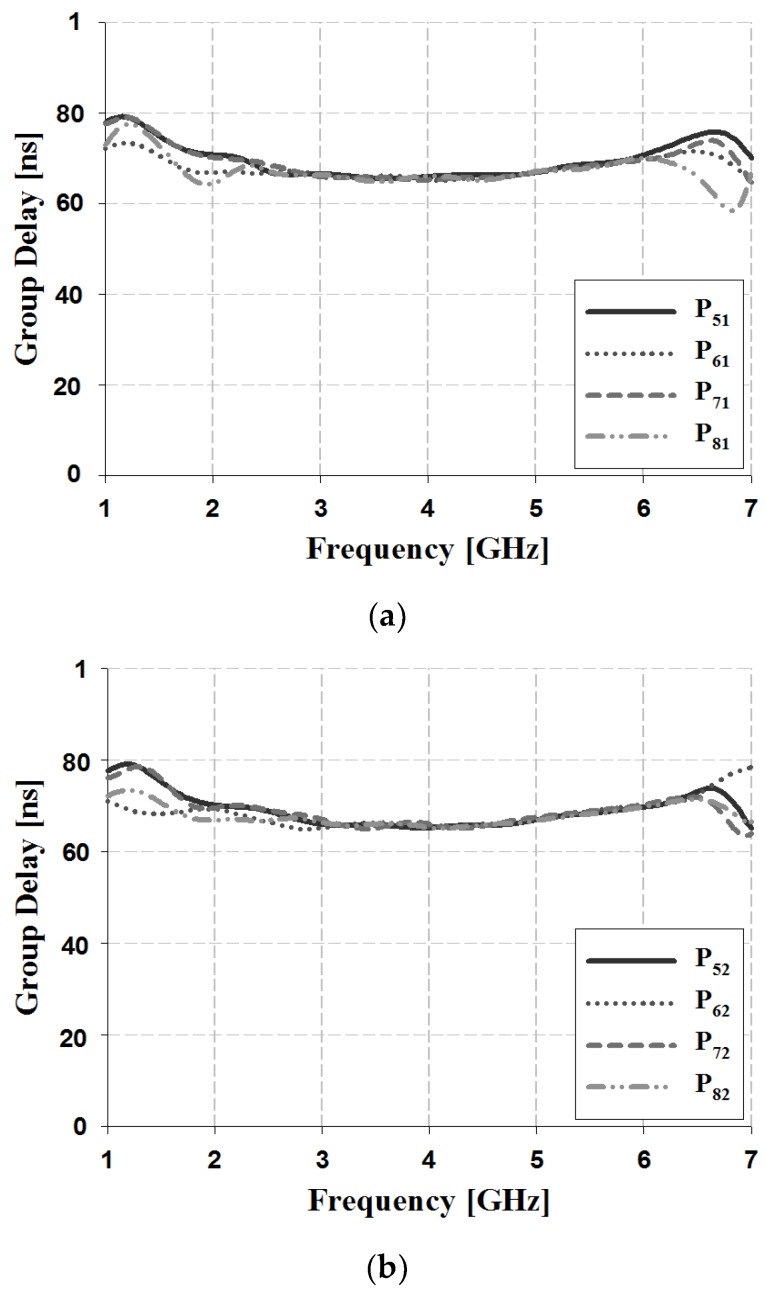
Group delay analysis of the proposed 4 × 4 Butler matrix (**a**) results of output ports for input Port 1, (**b**) results of output ports for input Port 2.

**Figure 23 sensors-19-03040-f023:**
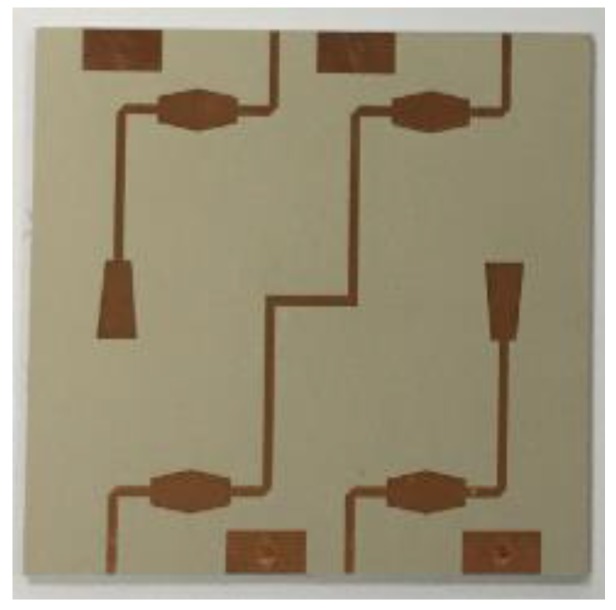
Photograph of fabricated 4 × 4 Butler matrix.

**Figure 24 sensors-19-03040-f024:**
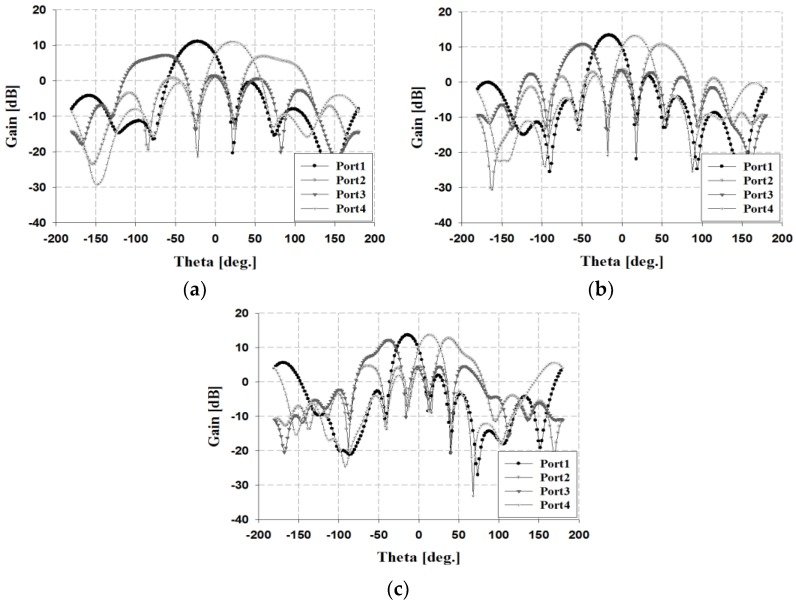
Simulation results of the fabricated phased array antenna at each frequency: (**a**) 3, (**b**) 4, and (**c**) 5 GHz.

**Figure 25 sensors-19-03040-f025:**
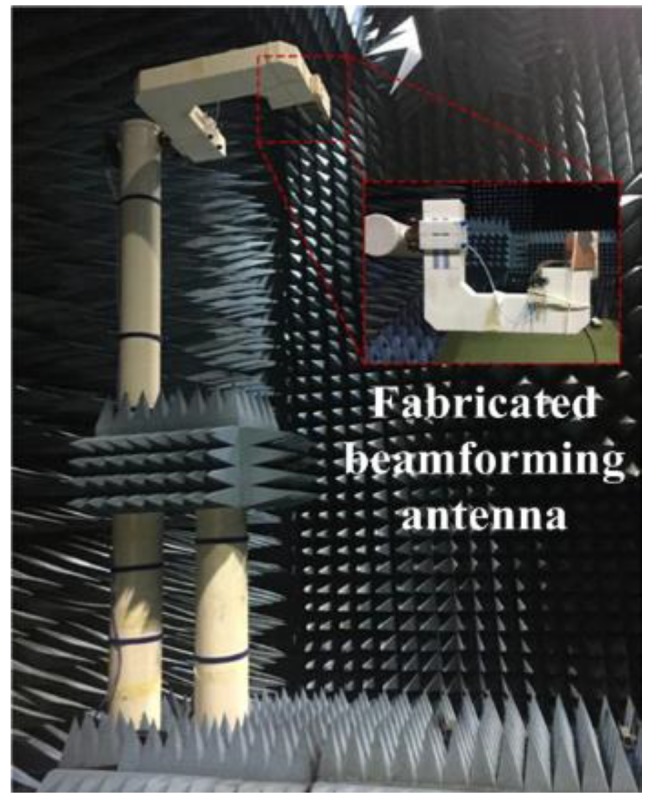
Measurement configuration of the fabricated beamforming antenna.

**Figure 26 sensors-19-03040-f026:**
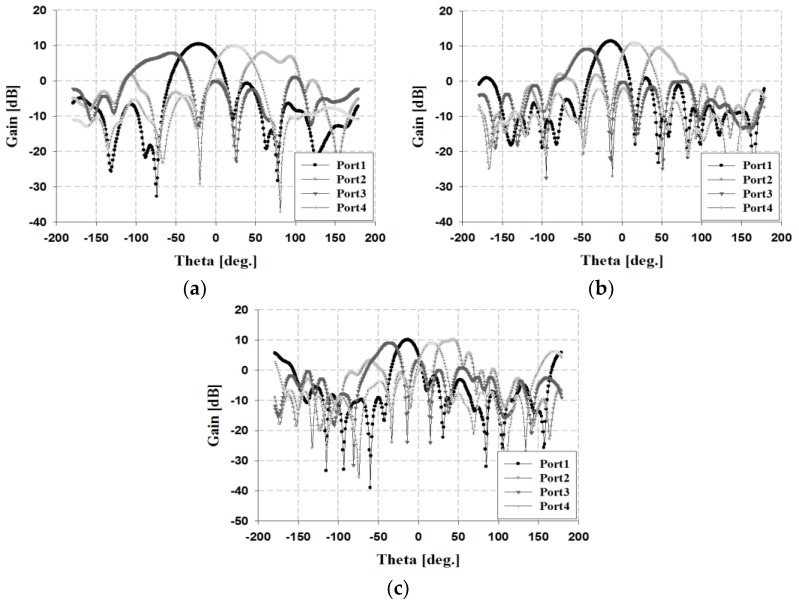
Measured results of the fabricated phased array antenna at each frequency: (**a**) 3, (**b**) 4, and (**c**) 5 GHz.

**Figure 27 sensors-19-03040-f027:**
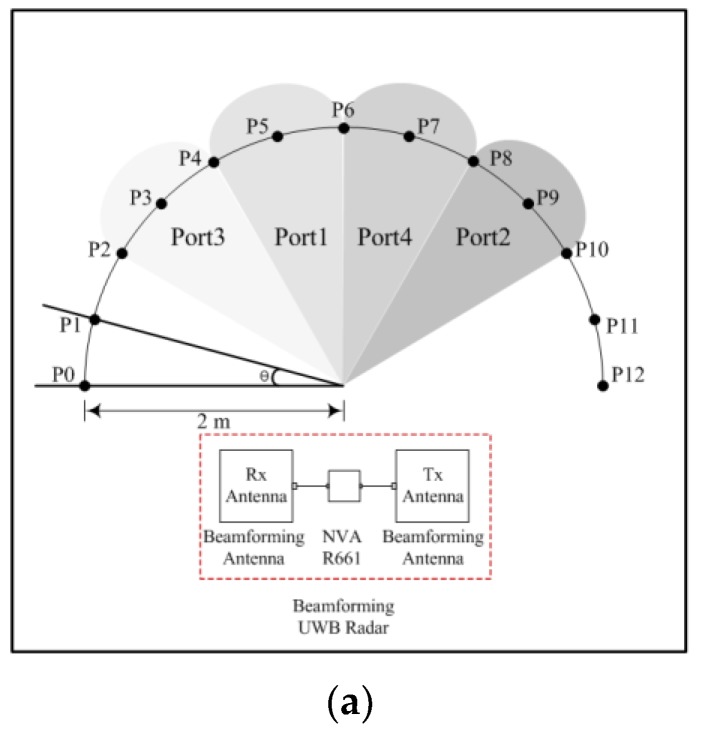
Proposed system: experiment environments (**a**) Configuration (**b**) #1 and (**c**) #2.

**Figure 28 sensors-19-03040-f028:**
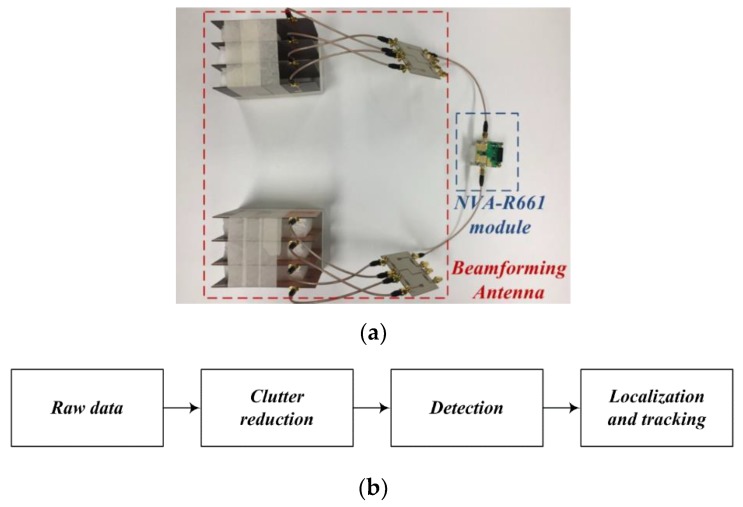
Configuration of the proposed radar system: (**a**) radar connected to beamforming antenna and NVA-R661 module and (**b**) signal processing.

**Figure 29 sensors-19-03040-f029:**
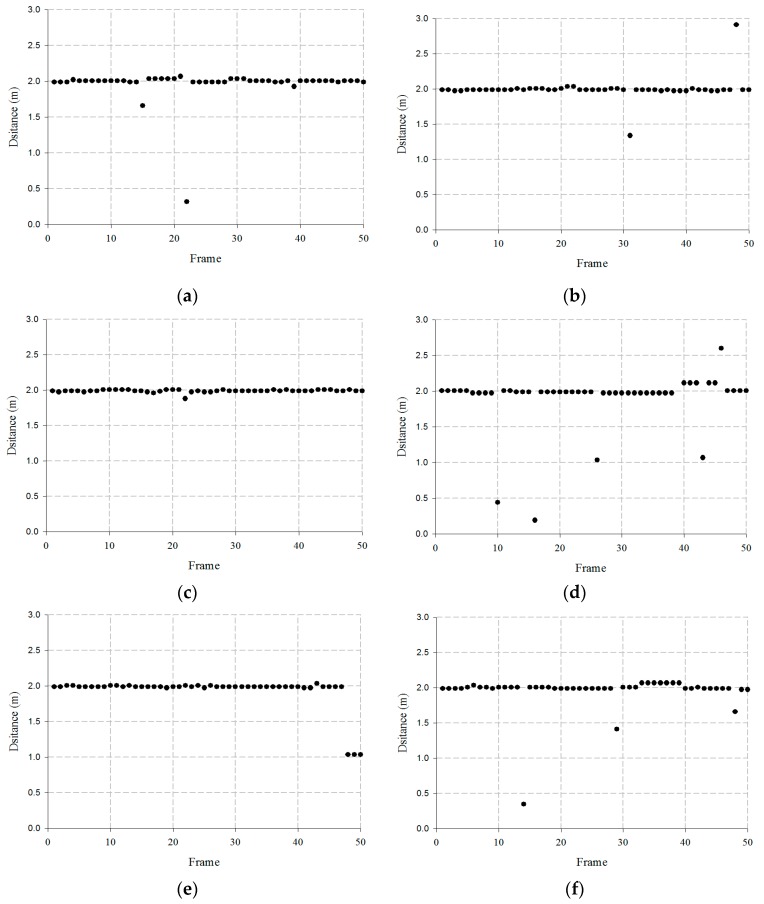
Measured results of single target using proposed beamforming antenna: (**a**) P2, (**b**) P3, (**c**) P4, (**d**) P4, (**e**) P5, (**f**) P6, (**g**) P6, (**h**) P7, (**i**) P8, (**j**) P8, (**k**) P9, and (**l**) P10.

**Table 1 sensors-19-03040-t001:** Comparisons of simulation and measurement results.

Parameters	Simulation Results			Measurement Results		
Impedance bandwidth	1.45–5.74 GHz			1.46–5.78 GHz		
Antenna Beamwidth	3 GHz	E	70°	3 GHz	E	72°
		H	117°		H	109°
	4 GHz	E	58°	4 GHz	E	40°
		H	96°		H	84°
	5 GHz	E	36°	5 GHz	E	29°
		H	72°		H	60°
Antenna gain	3 GHz	6.70 dBi		3 GHz	6.17 dBi	
	4 GHz	8.36 dBi		4 GHz	8.19 dBi	
	5 GHz	8.97 dBi		5 GHz	9.18 dBi	

**Table 2 sensors-19-03040-t002:** Simulation results of phase difference for 4 × 4 Butler matrix.

	Output Port	Phase [Deg.]				Phase Difference [Deg.]		
Input Port		Port 5	Port 6	Port 7	Port 8	P_5x_–P_6x_	P_6x_–P_7x_	P_7x_–P_8x_
Port 1	3 GHz	17	59	103	148	−42	−44	−45
	4 GHz	139	−177	−133	−88	−44	−44	−45
	5 GHz	−100	−56	−10	34	−44	−46	−44
Port 2	3 GHz	103	−29	−167	58	132	138	135
	4 GHz	−133	92	−44	−178	135	136	134
	5 GHz	−10	−147	76	−56	135	138	132

**Table 3 sensors-19-03040-t003:** Measurement results of phase difference for 4 × 4 Butler matrix.

	Output Port	Phase [Deg.]				Phase Difference [Deg.]		
Input Port		Port 5	Port 6	Port 7	Port 8	P_5x_–P_6x_	P_6x_–P_7x_	P_7x_–P_8x_
Port 1	3 GHz	−68	−25	20	62	−43	−45	−42
	4 GHz	23	66	116	155	−43	−50	−39
	5 GHz	115	157	−156	−113	−42	−47	−43
Port 2	3 GHz	19	−116	109	−30	135	135	139
	4 GHz	115	−25	−156	58	140	131	146
	5 GHz	−153	63	−62	156	144	125	142

**Table 4 sensors-19-03040-t004:** Comparisons of the proposed 4 × 4 beamforming antenna and reference antennas.

	Ref. [9]	Ref. [19]	Ref. [20]	This Work
Antenna Type	Beamforming antenna	Single antenna	Single antenna	Beamforming Antenna
Measurement Set-up	-	1D set-up	2D set-up	Set-up for Beamforming

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
