# Peer review of "Analysis of Beamforming Antenna for Practical Indoor Location-Tracking Application"

_sensors, 2019, doi:10.3390/s19143040_

Round 1
Reviewer 1 Report
The authors have synthesized a Butler matrix BFN for a 4 element UWB array. There are two main issues in the paper that do not allow its publication in present form.
1) Butler matrices are not suited for UWB systems, in these cases you should use a true-time-delay BFN. As it is possible to see from your results, the maxima of the scanned beams point in different directions for a variable frequency.
2) There are almost no information on the elaboration of the four port signals to achieve the target range detection of fig. 28. Why the detected distance is 2m when the target is not present?
Reviewer 2 Report
How about Figure x or Fig. x.? There are not corresponed each other in the expression in the inner text and figure description.
The proposed algorithm shall compare with the other methodology within recently five years journal papers to prove the paper contributions.
It’s not only to derive the mathematical expressions for the proposed method system but also implemented by applications for getting more comparisons.
Reviewer 3 Report
Although addressing an important issue and with some interesting results, I cannot recommend its publication in its present form. When revision the paper, the authors should have in mind the following issues:
- The writing needs a careful revision.
- It is not clear what is new and what comes from the literature.
- The performance results are not presented in a concise way.
Round 2
Reviewer 1 Report
The authors have answered my previous concerns.
Reviewer 3 Report
The authors improved substantially the paper.